# *Acacia senegal* Budmunchiamines as a Potential Adjuvant for Rejuvenating Phenicol Activities towards *Escherichia coli*-Resistant Strains

**DOI:** 10.3390/ijms24108790

**Published:** 2023-05-15

**Authors:** René Dofini Magnini, François Pedinielli, Julia Vergalli, Noufou Ouedraogo, Simon Remy, Adama Hilou, Jean-Michel Brunel, Jean-Marie Pagès, Anne Davin-Regli

**Affiliations:** 1UMR_MD1, U-1261, INSERM, SSA, IRBA, MCT, Faculté de Pharmacie, Université Aix-Marseille, 13385 Marseille, France; dofinirene@gmail.com (R.D.M.); julia.vergalli@univ-amu.fr (J.V.); bruneljm@yahoo.fr (J.-M.B.); anne-veronique.regli@univ-amu.fr (A.D.-R.); 2Laboratoire de Recherche-Développement de Phytomédicaments et Médicaments (LR-D/PM), IRSS, CNRST, Département MEPHATRA-PH, Ouagadougou 03 BP 7047, Burkina Faso; ouednouf@gmail.com; 3Laboratoire de Biochimie et de Chimie Appliquée (LABIOCA), Université Joseph Ki-Zerbo, Ouagadougou 03 BP 7021, Burkina Faso; hiloudio@gmail.com; 4Institut de Chimie Moléculaire de Reims, UMR CNRS 7312, Université Reims-Champagne-Ardenne, UFR Sciences, BP 1039, CEDEX 2, 51687 Reims, France; pedinielli@insa-toulouse.fr (F.P.); simon.remy@cnrs.fr (S.R.)

**Keywords:** *Acacia senegal*, phenicols, antibiotic resistance, efflux pumps, Budmunchiamines, adjuvant, natural metabolites, *Escherichia coli*, *Enterobacteriaceae*

## Abstract

The continuous emergence of bacterial resistance alters the activities of different antibiotic families and requires appropriate strategies to solve therapeutic impasses. Medicinal plants are an attractive source for researching alternative and original therapeutic molecules. In this study, the fractionation of natural extracts from *A. senegal* and the determination of antibacterial activities are associated with molecular networking and tandem mass spectrometry (MS/MS) data used to characterize active molecule(s). The activities of the combinations, which included various fractions plus an antibiotic, were investigated using the “chessboard” test. Bio-guided fractionation allowed the authors to obtain individually active or synergistic fractions with chloramphenicol activity. An LC-MS/MS analysis of the fraction of interest and molecular array reorganization showed that most identified compounds are Budmunchiamines (macrocyclic alkaloids). This study describes an interesting source of bioactive secondary metabolites structurally related to Budmunchiamines that are able to rejuvenate a significant chloramphenicol activity in strains that produce an AcrB efflux pump. They will pave the way for researching new active molecules for restoring the activity of antibiotics that are substrates of efflux pumps in enterobacterial-resistant strains.

## 1. Introduction

In recent last decades, antibiotic misuse and overuse have caused the re-emergence of a bacterial threat created by bacterial adaptation and the spread of resistance mechanisms. Consequently, during the same period, infectious bacterial diseases have become difficult to treat, and the prevalence of clinical therapeutic failures dramatically has increased due to the severe limitations of new antibacterial molecules [1,2]. Today, microbial resistance causes over 700,000 deaths annually worldwide and will cause 10 million by 2050 if nothing changes [3]. Among the various mechanisms involved in bacterial drug resistance, e.g., target mutations, antibiotic modifications, envelope alterations, etc., the control of a low internal concentration of antibiotics close to their target is one of the main mechanisms that contributes to resistance in Gram-negative bacteria [4]. Importantly, the microorganisms express efflux pumps that expel the antibiotic outside the bacteria earlier, before it acts on its target [4].

With the scarcity of new, active molecules available in clinics, research strategies to curb antibiotic resistance are actively being developed. In Africa, where the lack of antibiotics is particularly worrying, this aspect is actively developed in Burkina Faso, and 60–80% of the population relies on traditional medicine for primary health care [5]. Consequently, with the emergence of multi-resistant bacteria and the difficulty in treating infectious diseases, compounds obtained from medicinal plants are potential candidates for discovering original antimicrobial agents. One of the most promising means is studying new active compounds acting as adjuvant molecules which could enhance the activity of available antibiotics against bacterial strains exhibiting a high resistance level. A recent review presented the state of the art of research on the use of plant extracts to combat bacterial pathogens [6].

Previously, our studies showed that the hydroethanolic extract of the leaves of *Acacia senegal* had low levels of direct activity; however, in combination with phenicols, the extract rejuvenated the phenicols’ activity against resistant bacteria [7]. This antibiotic is a well-known substrate of Gram-negative bacteria efflux pumps such as RND pumps and is used to monitor the activity of efflux pumps in the presence of plant extracts [7,8]. Moreover, the hydroethanolic extract of *Acacia senegal* leaves (HEASG) was able to permeabilize the bacterial outer membrane and exhibited a relatively low level of toxicity [9]. Preliminary phytochemical studies have shown the presence of several groups of specialized metabolites, some of which present strong antibacterial potential [7]. It is important to characterize the chemical type of these metabolites active toward resistant bacteria, and our aims were first to fractionate HEASG and to evaluate the effect of the fractions individually and in combination with chloramphenicol on resistant bacterial strains. The supposed compounds that would be responsible for the direct and indirect antibacterial activity were identified using spectral analysis in a second step.

## 2. Results

### 2.1. Purification of the HEASG Extract by Flash Chromatography

Briefly, the purification of 9.5 g of *A. senegal* leaf extract via flash chromatography (Figure 1) yielded 60 fractions of 20 mL which were grouped into 7 fractions titled AS1 to AS7, as shown in Table 1.

The AS6 fraction was used for a second purification as it showed the best direct antibacterial activity and modulated the activities of chloramphenicol and florfenicol by increasing their activity against the selected bacteria. The purification via flash chromatography (Figure 1) of the AS6 fraction was carried out from 234.6 mg and allowed us to obtain 40 fractions of 20 mL, which were grouped into 10 fractions coded from AS6.1 to AS6.10. These fractions are summarized in Table 2, and their respective antibacterial activities are highlighted.

### 2.2. Effect of Extract Fractions on Direct and Adjuvant Antibacterial Assays

The bio-guided fractionation aimed at identifying two types of activities: a direct antimicrobial activity against resistant strains of *Escherichia coli* and *Klebsiella aerogenes*, represented by the minimum inhibitory concentrations (MICs, expressed in mg/L) and the adjuvant’s capability to modulate the chloramphenicol activity in *E. coli* strains to overexpress the AcrAB efflux pump. The fractionation of the hydroethanolic extract of *A. senegal* leaves generated seven subfractions. Table 3 presents the results of the direct activity of the 7 fractions from HEASG, and Table 4 presents those of the 10 sub-fractions of AS6. The MICs varied from 16 mg/L to the highest concentration of 128 mg/L. This bio-guided fractionation allowed us to highlight different aspects. On one hand, fractions with antibacterial activities, such as the AS6 fraction, were highlighted: this fraction presented a MIC of 16 mg/L against AG102 overexpressing the AcrAB efflux pump (which is not negligible since this activity is superior to the activity of chloramphenicol), in addition to a good modulating activity that may be due to a synergistic effect.

The activities of the subfractions obtained from AS6 are presented in Table 4. The best fractions inhibiting the growth of the *Escherichia coli* strains were AS6.1, AS6.6, AS6.7, AS6.8, and AS6.9, with MICs lower than those of chloramphenicol against AG102 overexpressing the efflux pump.

The adjuvant properties of the fractions are summarized in Table 5. The MIC of chloramphenicol was 8 mg/L in strain AG100, which expresses a basal level of the AcrAB efflux pump, and it was 64 mg/L in the multidrug-resistant strain AG102 overexpressing the AcrAB transporter. The susceptible strain AG100A, which is devoid of AcrAB, had a 1 mg/L MIC. Different concentrations of the fractions used did not alter the activity of chloramphenicol against the efflux minus AG100A. However, the combination of chloramphenicol and the AS6 fraction at a concentration of 4 mg/L reduced the MIC of the chloramphenicol from 64 to 4 mg/L in AG102 (overexpressing the efflux pump) and from 8 to 2 mg/L in AG100 (normal efflux), i.e., gains of 16 and 4, respectively. These results suggest that AS6 can potentialize the CHL activity in strains overproducing the AcrAB pump.

### 2.3. NMR Structure Determination

NMR spectroscopy was the starting tool that first proved the existence of a major product (Appendix A) [10,11,12,13,14]. However, NMR can easily be limited in identifying minor products. In such cases, the signals of less-abundant analytes are typically masked by the signals of metabolites present in larger quantities. This results in a high degree of overlap of proton resonances, especially in a spectral window of less than 4 ppm (Appendix A). The analytical challenge lies in the narrow range, which makes it difficult to assign the major alkaloid macrocycle based solely on the 1D ^1^H spectrum.

Moreover, when there is a large concentration ratio between entities, the receiver gain setting is not optimized for compounds at low concentrations, making it difficult to analyze compounds in low quantities. In these circumstances, SM is a major asset in making secondary structures detectable. A current alternative to overcoming the problem of signal overlap in NMR is to spread chemical shifts over a second dimension of nuclei other than 1H, such as the ^13^C and ^15^N nuclei. This approach to fraction analysis is illustrated in methanol-*d*_4_ using the homonuclear 2D experiments COSY ^1^H-^1^H, TOCSY ^1^H-^1^H, NOESY ^1^H-^1^H, and heteronuclear 2D HSQC ^1^H-^13^C, HMBC ^1^H-^13^C, HMBC ^1^H-^15^N, H2BC ^1^H-^13^C, and HSQC-TOCSY ^1^H-^13^C, providing lists of the chemical shifts ^1^H/^13^C and ^1^H/^15^N of the major Budmunchiamines.

The analysis of the NMR data revealed resonances that are compatible with the amide part (^13^C: δ 173.56 (C_2_); ^15^N: δ 121.4; ^1^H: δ 8.48 (1H, s, (H_1_))). The comparison of the HMBC ^1^H-^13^C/^1^H-^15^N spectra for the signals of the ketone C_2_, δ 173.56, and the NH, δ 121.4, constituted an anchor point for elucidating the macrocycle structure. Similar correlation tasks were observed with the resonances of the keto-methylene group (^13^C: δ 32.28 (C_3_); ^1^H: δ 2.69 (1H, dd, *J* = 17.10–8.22 Hz, (H_3_)), δ 2.94 (1H, dd, *J* = 17.08–2.90 Hz, (H_3_)), the amide methylene (^13^C: δ 34.97 (C_17_); ^1^H: δ 3.32 (1H, m, (H_17_)), δ 3.51 (1H, m, (H_17_))), and the CH_2_ (C_16_), which presents direct correlations with the amide methylene (C_17_), information confirmed utilizing the 2D H2BC (heteronuclear correlations ^2^*J* only) and 2D HSQC-TOCSY ^1^H-^13^C experiments. The presence of an amino-methylene (^13^C: δ 55.05 (C_4_); ^1^H: δ 3.63 (1H, m, (H_4_))) was confirmed by the last observable correlation of the ketone C_2_ on the HMBC spectrum as well as by the 2D H2BC experiment, which revealed the direct coupling with the ketone-methylene (C_3_) carbon and the first carbon of the aliphatic chain (C_1′_), constituting a bridge with the second NH, δ 51.7 (Appendix A). Further analysis of the ^1^H NMR data showed a triplet at δ 0.92 and a broad singlet at δ 1.31 characteristic of a straight-chain alkane. In the absence of sp or sp2 carbons in the NMR data for the major signals, it is evident that it is a monocyclic macrocycle. The NMR data also revealed resonances consistent with five amino-methylenes (^13^C: δ 41.69 (C_6_), δ 42.79 (C_8_), δ 45.06 (C_10_), δ 46.24 (C_13_), δ 44.39 (C_15_); ^1^H: δ 3.34 (1H, m, (H_6_)), δ 3.40 (1H, m, (H_6_)), δ 3.30 (1H, m, (H_8_)), δ 3.41 (1H, m, (H_8_)), δ 3.21 (1H, m, (H_10_)), δ 3.26 (1H, dt, *J* = 12.93–6.35 Hz (H_10_)), δ 3.16 (1H, m, (H_13_)), δ 3.22 (1H, m, (H_13_)), δ 3.15 (2H, m, (H_15_))), allowing for the determination of the location of the last two NH protons, δ 43.07 and 41.84 (Appendix A). These data suggest that the compound is a spermine-type alkaloid [12]. Although it can be difficult to differentiate the number of protons contained under the broad singlet at δ 1.31 and thus define the length of the alkyl chain, it is nevertheless possible to deduce the missing carbons by integrating each carbon peak for one in the ^13^C spectrum. In this way, 28 carbons were found, including 3 peaks that integrated for 10 carbons (^13^C: 29.05 (3C: C_3′_, C_4′_, C_13′_), 29.22 (1C), 29.36 (6C including C_11′_)) and had correlations only with the CH_2_ of the broad singlet that overlaps at δ 1.31 in the 2D HSQC ^1^H-^13^C spectrum (Appendix A). Thus, the broad singlet contains 26 protons carried by 13 carbons (C_2′_-C_14′_). A comparison of the NMR data with known spermine-type alkaloids found in the *Albizia lebbek*, Budmunchiamines L1-L6 [14,15,16], indicated that the main difference lies in the length of the side chain of the saturated alkane. Therefore, the structure found is a Budmunchiamine L5 (Appendix A) with a molecular formula of C_28_H_58_N_4_O, as Ovenden et al. [16] suggested.

### 2.4. Structure Annotation of LC-MS/MS

Based on the results obtained in the microbiology (MICs) and NMR analyses with the bio-guided fractionation, the group of molecules forming the molecular network highlighted in Figure 2 was composed of ions exclusively detected in the fraction ASG, regrouping the fractions AS5 and AS6.1, AS6.6, AS6.7, AS6.8, and AS6.9. The molecular network was constructed by mapping the fragment spectra to the GNPS reference spectra databases (http://gnps.ucsd.edu, accessed on 13 April 2023). The annotated molecule families were also alkaloids, terpenoids, fatty acids, amino acids, peptides, etc. The size of the node is a function of the mass intensity. One of these nodes represented predominantly was noted as a Budmunchiamine (alkaloids-like macrocyclic polyamine), as illustrated in Figure 3 by node B in the network and its analogs (nodes A, C, D, and E), based on the molecular formula deduced from the protonated ions at *m/z*. The characteristics of the nodes are represented in Figure 3.

**Node A**: an HRESIMS ([M+H]^+^) analysis showed a protonated molecular ion peak (*m/z*) at 439.4377 (calculated 438.4377), corresponding to the following formula, C_26_H_54_N_4_O, with a retention time of 5.6967min (Figure 3a). It fits in well with Budmunchiamine L4 [17,18], which was previously isolated from *Albizia amara* and, to our knowledge, annotated for the first time in a plant extract of the genus *Acacia* and of the species *A. Senegal.*

**Node B:** a positive-mode HRESIMS ([M+H]^+^) analysis showed a molecular ion peak (*m/z*) at 467.4687 (calculated at 466.4687) that corresponds to the formula C_28_H_58_N_4_O, with a retention time of 6.7768 min (Figure 3b). It was noted as Budmunchiamine L5 or G [18,19], previously isolated from *Albizia amara*. The molecular peak is identified for the first time from the genus *Acacia* and the species *A. senegal*.

**Node C**: an HRESIMS ([M+H]^+^) analysis showed a fragment ion peak (*m/z*) at 493.4848 (calculated 492.4848), corresponding to the formula C_30_H_60_N_4_O, and the retention time was estimated to be 7.3425 min (Figure 3c). The compound was annotated as an analog of Budmunchiamines L4 and L5, with a double bond at two units of the long-chain terminal carbon. This molecule is also described for the first time in the genus *Acacia* and the species *A. senegal*.

**Node D**: an HRESIMS ([M+H]^+^) analysis of node 4 showed a fragment ion peak (*m/z*) at 495.5007 (calculated 494.5007), corresponding to the formula C_30_H_62_N_4_O. The retention time was estimated to be 7.8909 min (Figure 3d). This node was annotated as an analog of Budmunchiamines A, B, and C, previously isolated from *Albizia amara* [18], and an analog of 9-Normethylbudmunchiamine K, isolated from *Albizia gummifera* [17]. This type of compound is also described for the first time in the genus *Acacia* and the species *A. senegal*.

**Node E**: an analysis of the high-resolution EI ([M+H]^+^) mass spectrum of node 5 shows a significant ion peak (*m/z*) at 537.5473 (calculated at 536.5473) which corresponds to the formula C_33_H_68_N_4_O (Figure 3e). The retention time was estimated to be 9.5303 min. This compound was also annotated as an analog of node 4 (a macrocycle of 42 more mass units) and the Budmunchiamines A, B, and C mentioned above. Like the previous ones, this molecule is described for the first time in the genus *Acacia* and in the species *A. senegal*.

## 3. Discussion

Based on the spectral analysis and comparison with related compounds, the fraction extract of the *Acacia senegal* leaves allowed for the annotation of five alkaloidal macrocyclic polyamine compounds. The first two already known from the literature are Budmunchiamine L4 (node A: C_26_H_54_N_4_O) and Budmunchiamine L5 (node B: C_28_H_58_N_4_O). Budmunchiamine 3 (node C: C_30_H_60_N_4_O) is thought to be a new analog of Budmunchiamines L4 and L5 and to our knowledge, it has never been described. As for nodes D (C_30_H_62_N_4_O) and E (C_33_H_68_N_4_O), these Budmunchiamines would be new analogs of Budmunchiamines A, B, and C. It is important to note that the species in which Budmunchiamines (*Albizia amara, Albizia gummifera* etc.) were first identified belong to the same family as *Acacia senegal*, the family Fabaceae. Previous work indicated that Budmunchiamines (A, B, and C) were isolated from *Albizia amara* and exhibited noticeable biological activity by interacting with calf thymus DNA. They inhibit the catalytic activities of DNA polymerase, RNA polymerase, and HIV-1 reverse transcriptase and exhibit an anti-cyclooxygenase activity. Considering the role played by DNA polymerase in DNA replication, its inhibition will lead to the arrest of cell multiplication. Interestingly, a mixture of Budmunchiamines presenting a bactericidal activity against *S. Typhimurium* was also reported [20]. Moreover, Budmunchiamine A, isolated from *A. amara*, showed a concentration-dependent bactericidal activity against Gram-positive (*S. aureus, S. faecalis*) and Gram-negative bacteria (*E. coli, K. pneumonia, P. aeruginosa*) and an antifungal activity against *C. albicans* and *C. neoformans*. In addition, it exhibited moderate antioxidant activity with an IC_50_ value of 400 µg/mL, determined via the DPPH method, and a 67.8% inhibition of carotene/linoleic acid at 1000 µg/mL [21]. Budmunchiamine A significantly inhibited the growth of a wide range of seed-borne corn fungi, including aflatoxin from *Aspergillus flavus*, and increased the seedling vigor index in vivo at a lower concentration (1.0 g/kg) than the pithecolobin molecule (2.0 g/kg) [22]. Samoylenko et al. [23] isolated four Budmunchiamine analogs from *Albizia schimperiana*.

From the assays regarding antibacterial and anticancer activities, some compounds showed significant effects, and others had moderate effects. The authors noted that a hydroxyl group on the side chain decreased the antibacterial and cytotoxic activities. A similar trend of decreased toxicity in mammalian cell lines was observed for Budmunchiamines bearing carboxyl groups on the side chain [19,20].

The structures of the analogs observed in the fraction do not have hydroxyl or carboxyl groups on the side chain. These results demonstrate that the potential of Budmunchiamines is of great importance given their activities on bacteria that present a high level of antibiotic resistance. Polyamines such as spermine and squalamine have been shown to inhibit wall synthesis in Gram-negative bacteria [24]. Similarly, Borselli et al. (2017) demonstrated that Motuporamin C (marine macrocyclic alkaloid) and its derivatives had excellent antimicrobial activities against many species, including *K. aerogenes* EA289, which overexpresses the AcrAB-TolC efflux pump [25]. The results showed that Motuporamines permeabilize or disrupt the outer membrane of Gram-negative bacteria. They can also disrupt the proton gradient, de-exciting the efflux pump in *K. aerogenes* EA289, and depolarize the bacterial inner membrane. Previous studies have also shown that the combination of chloramphenicol with polyamines enhances the antibacterial activity of the antibiotic by preferentially absorbing spermidine, thereby increasing the internalization of the antibiotic and interfering with protein synthesis via polyamine metabolism, which leads to a decrease in protein and polyamine contents in *E. coli* [26].

Interestingly, this study indicates that the Budmunchiamines analogs from *A. senegal* leaves identified here have significant activity against AcrB efflux and, taking into account previous observations [6], present a selectivity for the phenicol site(s) located inside the AcrB transporter. This original observation is important with respect to the recent data stating the relative selectivity of the AcrB pump for antibiotics belonging to the fluoroquinolone group [27]. The authors report that fluoroquinolones are more efficiently expelled than other antibiotics; this difference in the expel flux is associated with the chemical structure of the substrate, indicating the key role of specific pharmacophores in the recognition of and binding to specific amino acid residues present in the internal AcrB pocket.

## 4. Materials and Methods

### 4.1. Plant Collection

Plant samples were collected during June and August 2018 in Saaba, Gonsé, about twenty kilometers from Ouagadougou (Burkina Faso). The plants were identified by Pr A. Ouedraogo of the Botany Section, University Joseph KI-ZERBO of Ouagadougou, voucher specimens were deposited under reference numbers 6896/17257, and the GPS coordinates were listed as N 12°26.611 W 001°20.991. The leaves were dried under ventilation in the dark for 21 days. The samples were then ground into a fine powder using a blade mill (Gladiator Est., 1931 Type B.N. 1 Mach 40,461 1083). The powders were placed in freezer bags and stored at 4 °C for further use.

### 4.2. Hydroethanolic Extracts (HE)

Extracts were prepared by soaking 100 g of powdered *A. senegal* leaves in 500 mL of ether petroleum for 24 h. The residue was filtered through Whatman filter paper N°1, and the solids were dried and soaked in 1 L of 70% ethanol (*v*/*v*) overnight with shaking. Part of the solvent was removed under a vacuum, and the resulting paste was freeze-dried. This hydroethanolic extract was stored at 4 °C until use [7].

### 4.3. Flash Chromatography Purification

The crude sample (9.5 g dissolved in 3 mL of methanol) was purified on a CombiFlash system (Teledyne Isco, Lincoln, NE, USA) using a silica Redi SEP RF stationary phase (400 mg–80 g, Flash column; CV125 mL). Chromatographic separation was performed using a flow rate of 18 mL/min. The mobile phase was composed of solvent A1: petroleum ether, solvent A2: ethyl acetate, solvent B1: methanol, and solvent B2: 7/3/1 (CH_2_Cl_2_/MeOH/NH_4_OH), using a stepwise gradient elution type. The signal was monitored at 254 nm and 280 nm. According to staining, the fractions were collected and pooled into seven fractions.

### 4.4. Bacterial Strains

The phenotypic and/or genotypic characteristics of the strains used in this study are presented in Appendix A. *Escherichia coli* is the parental *E. coli* K-12 porin + and, basal efflux; *E. coli* AG100A is kanamycin-resistant derivatives with AcrAB deleted and is hyper-susceptible to chloramphenicol, tetracycline, ampicillin, and nalidixic acid; AG102, which overproduces the AcrAB efflux pump, is an AG100 derivative. *Klebsiella aerogenes*, the ATCC 15,038 strain, and resistant strains (ATCC derivative strain CM65 and resistant isolates Ea 289 and 298) are described in Appendix A. The bacteria were routinely grown at 37 °C on Luria–Bertani agar (LB).

### 4.5. Bacterial Susceptibility Determinations

The microdilution method determined direct antibacterial activity as described in [28,29]. Each fraction was analyzed using a two-fold dilution series prepared in dimethyl sulfoxide (DMSO) (1%). Microwells containing 100 µL of fraction dilutions were inoculated with 100 mL of a cell suspension prepared by diluting an overnight culture in twice-concentrated MHB to obtain viable counts of ca. 5 × 10^5^ CFU/mL at a final DMSO concentration of 0.5%, a concentration at which no significant effect was observed on the bacteria tested as a control under these conditions. Then, the inoculated microplates were incubated without shaking at 37 °C for 18 h. The samples’ minimal inhibitory concentrations (MICs) were observed after adding 40 µL of 0.2 mg/mL iodonitrotetrazolium chloride (INT). The MIC was defined as the lowest fraction concentration that inhibited the test strain’s visible growth. Experiments were carried out in triplicates. Chloramphenicol, an antibiotic well-described as an efflux substrate, was used to follow the activity of the efflux pump in the absence or presence of plant extracts [7].

### 4.6. Combination with Antibiotics

Our objective was to evaluate if the extracts were able to restore the activity of chloramphenicol against resistant bacteria. Thus, any direct effect of the extract needed to be avoided in order to be able to work in combination and to really detect the action of the antibiotic [8]. The activities of the various combinations, antibiotics, and fractions from the leaf extracts of *A. senegal* were determined via the checkerboard assay [29]. Serial dilutions of two antimicrobial agents were mixed so that each row (and column) contained a fixed concentration of the first agent (extract) and an increasing concentration of the second (antibiotic). The concentrations of the fractions were distributed vertically from 1 to 512 mg/L, and the antibiotic was dispatched from 1 to 1024 mg/L horizontally, according to the sensibility of each bacterium. Thus, 190 µL of a fresh bacterial suspension prepared in MH2 broth (5 × 10^5^ CFU/mL) was added to each well. The first column was used to determine the MIC of the extract alone. The microplates were closed and incubated for 18 h at 37 °C. The MICs of the samples were observed after adding 40 µL of 0.2 mg/mL p-iodonitrotetrazolium chloride (INT). MIC values were recorded as the lowest sample concentration that inhibited bacterial growth.

### 4.7. Ultra-Performance Liquid Chromatography Coupled to Mass Spectrometry Analysis

The sample was solubilized in methanol (MS-grade) at a concentration of 100 µg/mL and sonicated for 10 min. LC-MS/MS analyses were performed using equipment including an Acquity UPLC H-Class chain (Waters, Manchester, UK) coupled to an Equity UPLC PDA UV detector (Waters) and a Synapt G2-Si mass spectrometer (Waters) equipped with a time-of-volume analyzer. Analytical separation was performed using a, Uptisphere Strategy C18-HQ 2.2 µM, 150 × 2.1 mm analytical C18 column with a flow rate of 0.5 mL/min and a linear gradient from 10% B (A: H_2_O + 0.1% formic acid, solvent B: MeCN + 0.1% formic acid) to 100% B over 15 min. The wavelength range of the UV detector was 210–400 nm. The electrospray source was set as follows: positive mode; source temperature at 150 °C; capillary voltage at 3 kV; desolvation gas temperature at 600 °C; nebulizer pressure at 6 bar; desolvation gas flow rate at 700 L/h. MS scans were performed in full-scan mode from *m/z* 100 to 1500 (scan time: 0.1 s) with a resolution of 40,000 (FWHM). A scan of MS1 was followed by MS2 scans of the three most intense ions above a threshold of 3000 counts (exclusion window: 3 s). The selected parent ions were fragmented using the energy ramp with the following specifications: low mass start, 20 eV; low mass end; 30 eV; high mass start, 40 eV; high mass end, 70 eV. Leucine-enkephalin (1 ng/µL) was used as a reference mass via a lock spray interface at a flow rate of 10 μL/min for tracking in positive-ion mode ([M + H]^+^ = 556.2771). The MS data acquisitions were performed using Mass Lynx 4.1 software (Waters).

### 4.8. Molecular Networking

The MS/MS data were converted from files (.raw) to. mzXML files using M.S. Convert software (http://www.proteowizard.sourceforge.net, accessed on 13 April 2023) and imported and processed using Sirius 4.4.8 with the following module: SIRIUS molecular formula identification, ZODIAC, CSI: Finger ID fingerprint prediction and structure database search and CANOPUS compound class prediction (see parameters configuration in Appendix A). The .mgf file and the feature quantification table were exported using the FBMN export module of Sirius and then submitted to the Global Natural Product Social Molecular Networking (GNPS) platform (http://gnps.ucsd.edu, accessed on 13 April 2023) [30]. The data were filtered by removing all MS/MS peaks within ±17 Da of the precursor *m/z*. The data were then clustered with MS-Cluster with a parent mass tolerance of 2 Da and an MS/MS fragment ion tolerance of 0.5 Da to create consensus spectra.

Further, consensus spectra that contained less than two spectra were discarded. A network was then created in which the edges were filtered to achieve a cosine score above 0.7 and more than 6 matched peaks. Further edges between two nodes were kept in the network only if each of the nodes appeared in the other’s respective top ten most similar nodes. The spectra in the network were then searched against GNPS spectral libraries. The library spectra were filtered in the same manner as the input data. The data were then imported into Cytoscape v3.7.1 and visualized as a network of nodes and edges. The most consistent structure candidate provided by SIRIUS was depicted on each node using the chemviz2 plugin in Cytoscape. Data are available at https://gnps.ucsd.edu/ProteoSAFe/status.jsp?task=d9105ad8612b4927a707d7513ff82cf5, accessed on 13 April 2023.

### 4.9. NMR

All NMR experiments on the sample mixture were performed at 298 K in methanol-*d*_4_ on a Bruker Advance AVIII800 HD console NMR spectrometer equipped with a 5 mm TCI (^1^H/^13^C/^15^N/^31^P/^2^H) cryoprobe using the Bruker TOPSPIN Software (TopSpin 3.2., Rheinstetten, Germany). The temperature was controlled by a Bruker variable temperature (BSVT) unit supplied with chilled air produced by a Bruker cooling unit (BCU-Xtreme). Spectra were calibrated in methanol-*d*_4_ so that the proton resonance was observed at δ 4.78 ppm.

#### Statistics

The antibacterial assays were repeated three times on separate days, and similar results were obtained in each experiment. An Excel spreadsheet was used to create and format the tables to group and analyze the data.

## 5. Conclusions

These studies show that *Acacia senegal,* in addition to the molecules already isolated, exhibits compounds, namely, Budmunchiamines, that had not been previously mentioned in the species or even in the genus. We characterized five of them in the fraction of interest; two had already been described in *Albizia amara*, the Budmunchiamines L4 and L5, and three other analogs had never been described. Their structures were established from the GNPS database using the molecular network. As for their biological properties, it is reported that some Budmunchiamines have antibacterial effects, especially the Budmunchiamine A analog of node E. These molecules can selectively block the efflux of phenicol compounds, restoring antibacterial activity in strains overexpressing the AcrAB efflux pump, a pump reported in various clinical isolates that contributes to antibiotic resistance. Our study molecularly defines the structure of active molecules and allows us to scientifically validate the use of a traditional plant. Moreover, this supports the interest in the framework of a future medicinal/therapeutic antibacterial usefulness as an alternate way to combat multidrug-resistant bacteria in particular. Future studies will be focused on evaluating the potentiality of such derivatives in vivo by determining the cytotoxicity of these compounds and establishing the mechanism of action involved.

## Figures and Tables

**Figure 1 ijms-24-08790-f001:**
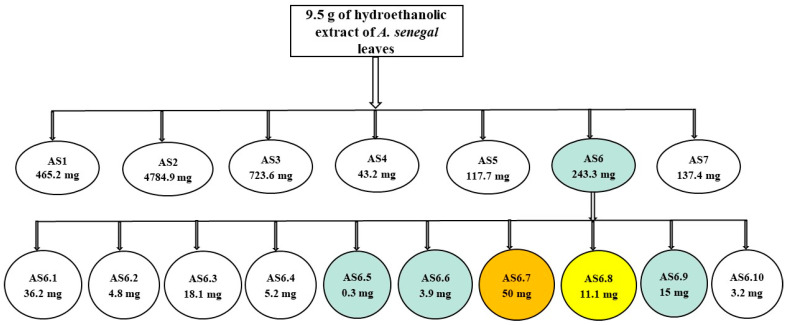
Summary of bio-guided fractionation. The colored part of the first row is the fraction of interest which was fractionated in a second time to obtain the sub-fractions and after the biological evaluation, the colored sub-fractions are those with which we obtained an activity.

**Figure 2 ijms-24-08790-f002:**
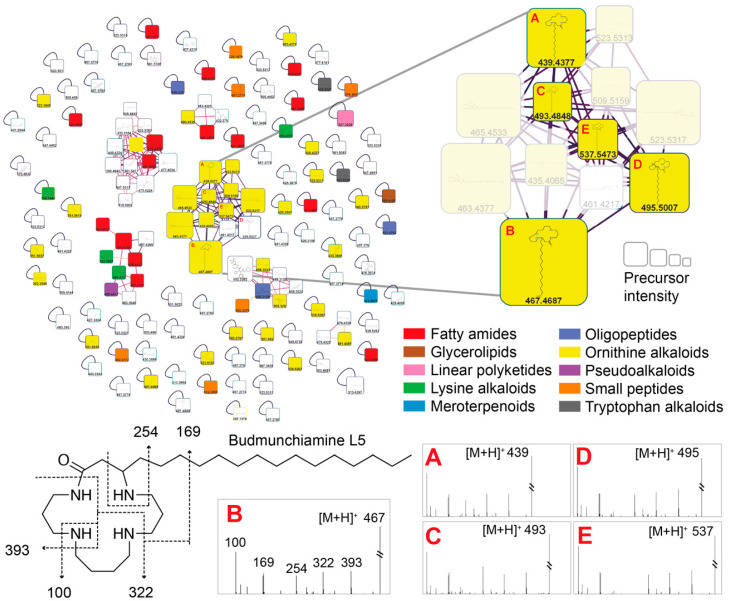
Molecular network of the fraction ASG. The colors represent the chemical class (NP classifier) of each feature’s annotation, and each node’s size is proportional to the precursor intensity. Experimental MS/MS spectra of features A–E annotated as Budmunchiamine derivatives and putative fragmentation pathway of Budmunchiamine L5.

**Figure 3 ijms-24-08790-f003:**
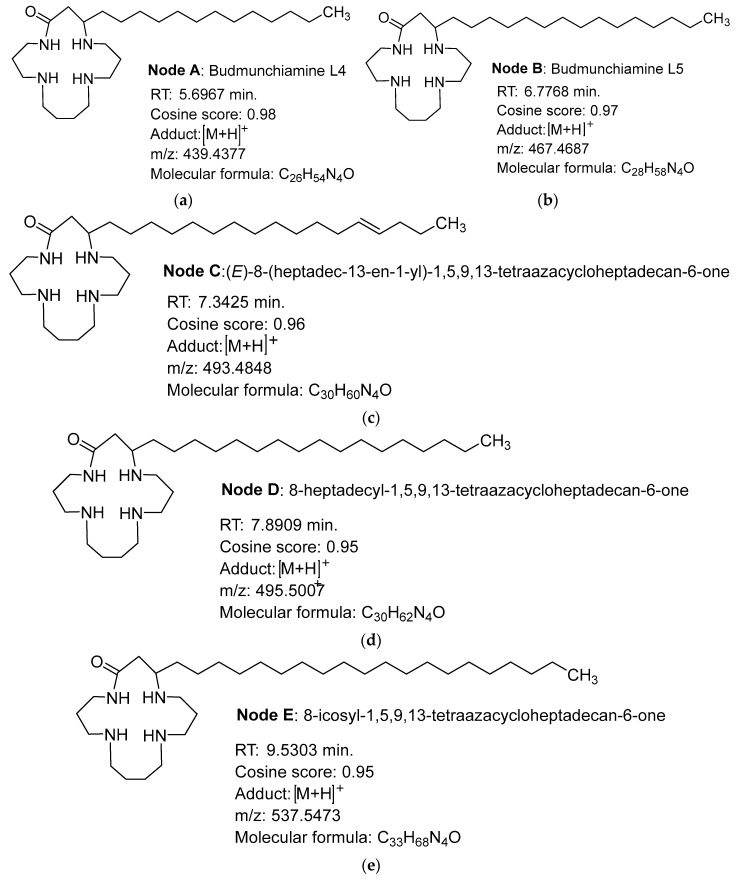
Structures of the Budmunchiamines annotated from the ASG fraction. (**a**–**e**) forming Figure 3 refer to the molecular structure of the Budmunchiamines annotated from the spectral analysis with certain parameters. 3a = Node A, Budmunchiamine L4; 3b = Node B, Budmunchiamine L5; 3c = Node C, analogues of Budmunchiamines L4 and L5; 3d = Node D, analogues of A, B and C and 3 = Node E, also analogue of Budmunchiamines A, B; and C.

**Table 1 ijms-24-08790-t001:** Summary of HEASG fractionation yields.

Fraction	Weight (mg)	Yield (%)
AS1 (F1–F17)	465	5
AS2 (F18–F33)	4785	50
AS3 (F34–F38)	724	8
AS4 (F39–F45)	43	0.5
AS5 (F46–F50)	118	1
AS6 (F51–F55)	243	3
AS7 (F56–F60)	137	2

**Table 2 ijms-24-08790-t002:** Summary of the fractionation of AS6.

Fractions	Weight (mg)	Yield (%)
AS6.1 (F1–F19)	36	15
AS6.2 (F20–F23)	5	2
AS6.3 (F24–F25)	18	8
AS6.4 (F26–F27)	5	2
AS6.5 (F28–F29)	1	1
AS6.6 (F30–F33)	4	2
AS6.7 (F34–F35)	50	21
AS6.8 (F36–F37)	11	5
AS6.9 (F38–F39)	15	6
AS6.10 (F40)	3	2

**Table 3 ijms-24-08790-t003:** Minimum inhibitory concentrations of HEASG fractions obtained (MIC in mg/L).

	*E. coli*			*K. aerogenes*			
Fractions	AG100	AG100A	AG102	Ea 289	Ea 298	EaATCC 15038	CM 64
**AS1**	>512	>512	>512	>512	>512	>512	>512
**AS2**	>512	>512	>512	>512	>512	>512	>512
**AS3**	>512	>512	>512	>512	>512	>512	>512
**AS4**	64	64	64	128	32	64	128
**AS5**	32	16	64	64	32	32	64
**AS6**	16	16	16	32	16	16	64
**AS7**	64	32	64	128	64	64	256

MIC: Minimum inhibitory concentration.

**Table 4 ijms-24-08790-t004:** Minimum inhibitory concentrations of AS6 subfractions.

*E. coli* MIC (mg/L)			
Fractions	AG100	AG100A	AG102
**AS6.1**	16	16	32
**AS6.2**	>64	>64	>64
**AS6.3**	>64	>64	>64
**AS6.4**	>64	64	>64
**AS6.5**	64	64	>64
**AS6.6**	16	16	16
**AS6.7**	8	16	16
**AS6.8**	8	16	8
**AS6.9**	16	16	16
AS6.10	>64	>64	>64

MIC: Minimum inhibitory concentration.

**Table 5 ijms-24-08790-t005:** Combination of AS5, AS6, and AS7 fractions with chloramphenicol (gain with minimal AS concentration).

Strains	ATBs	Fractions	Combination with AS5 at Conc		
*E. coli*	MIC CHL	MIC AS5	16	8	4
AG100	8	16	2	2	4
AG100A	1	16	-	1	1
AG102	64	32	8	16	32
			**Combination with AS6 at Conc**		
Strains	MIC CHL	MIC AS6	16	8	4
AG100	8	16	-	1	2 (4)
AG100A	1	16	-	1	1
AG102	64	16	-	2	4 (16)
			**Combination with AS7 at Conc**		
Strains	MIC CHL	MIC AS7	16	8	4
AG100	8	32	1	2	4
AG100A	1	32	1	1	1
AG102	64	32	4	8	32

MIC: minimum inhibitory concentration; Conc: concentration; CHL: chloramphenicol; ATBs: antibiotics.

## Data Availability

All data for this article are presented in the manuscript or provided as Appendix A.

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
