# Peer review of "Acacia senegal* Budmunchiamines as a Potential Adjuvant for Rejuvenating Phenicol Activities towards *Escherichia coli*-Resistant Strains"

_ijms, 2023, doi:10.3390/ijms24108790_

Round 1
Reviewer 1 Report
The topic of the manuscript (ijms-2373162-v1) is of interest because natural product plays a vital role in our life. As we know, the continuous emergence of bacterial resistance alters the activity of different antibiotic families and requires appropriate strategies to solve therapeutic impasses. The manuscript describes in general interesting laboratory work, which could be useful for other researchers and/or industry working on the topic. In the study, the fractionation of natural extracts from A. senegal and the determination of antibacterial activities are associated with molecular networking and tandem mass spectrometry data to characterize active molecules. The results indicated that A. senegal is a potential source of against antibiotic resistance. Regarding results, some concepts should be carefully revised. There are many problems that have not been explained clearly. 1) Whether the extracts show bacteriostatic or bactericidal effects? 2) What is the structure-activity relationship that works on which multidrug-resistant bacteria?
Author Response
Response to Referee 1 in blue
1) Whether the extracts show bacteriostatic or bactericidal effects?
In fact, our objective is to evaluate if the extracts are able to restore the activity of chloramphenicol against resistant bacteria. Thus, any direct effect of the extract must be avoided in order to be able to work in combination and, to be able to study only the action of the antibiotic.
At high concentration of extracts alone, the extracts show a bactericidal effect (as previously reported in ref 7, 9), no bactericidal or bacteriostatic effect was observed at concentration used during combination assay (see results obtained with efflux minus strain).
2) What is the structure-activity relationship that works on which multidrug-resistant bacteria?
Taking into account the similarities observed with Budmunchiamines, we mention in Discussion that Budmunchiamines analogs from A. Senegal show a significant restoring activity of chloramphenicol only in strain expressing AcrB efflux.
>>Taking into account the comments, we have added some modifications in the new version (yellow underlined)
Reviewer 2 Report
The title does not clearly indicate the work described in the manuscript.
The abstract does not provide outline of the work carried out and it has to be modified.
The background for the study given in the introduction is not clear. The authors write that adjuvant molecules (from natural products?) to increase antibiotic sensitivity and in the very next sentence, they write about Burkina Faso and then how natural products can be used as antimicrobials!
Why chloramphenicol was used? This antibiotic is rarely used because of its toxicity
In the methodology, only E. coli mentioned while in the results, authors have written that Klebsiella aerogenes was also used to test antibacterial effects.
Minor grammatical errors should be corrected.
Author Response
Responses to Referee 2 in blue
1. The title does not clearly indicate the work described in the manuscript.
>Title has been corrected
2. The abstract does not provide outline of the work carried out and it has to be modified.
> Abstract has been corrected
3. The background for the study given in the introduction is not clear.
> Introduction has been improved
4. The authors write that adjuvant molecules (from natural products?) to increase antibiotic sensitivity and in the very next sentence, they write about Burkina Faso and then how natural products can be used as antimicrobials!
> Text has been changed
5. Why chloramphenicol was used? This antibiotic is rarely used because of its toxicity
> This antibiotic is a well-known substrate of Gram-negative bacteria efflux pumps such as RND pumps and has been previously used to evaluate the activity of plant extracts (ref 7,9) (see the text added in Introduction). In addition, although Phenicols exhibit a toxicity that seriously compromises its use in human medicine, in meningitis caused by Haemophilus influenzae or Neisseria meningitidis when penicillin cannot be used (resistance, allergy), it diffuses easily into blood and tissues to eliminate pathogens (ref 7).
6. In the methodology, only E. coli mentioned while in the results, authors have written that Klebsiella aerogenes was also used to test antibacterial effects.
> Text has been corrected
7. Comments on the Quality of English Language
Minor grammatical errors should be corrected.
>Grammatical and typographical errors have been corrected along the manuscript (text, tables, figures, etc)
Round 2
Reviewer 2 Report
Nil